# Recent Developments in Rice Molecular Breeding for Tolerance to Heavy Metal Toxicity

Zulqarnain Haider [1,†], Irshan Ahmad [1,†], Samta Zia [2] and Yinbo Gan [1,*]

1 Department of Crop Genetics and Breeding, College of Agriculture and Biotechnology, Zhejiang University, Hangzhou 310058, China
2 College of Plant Science and Technology, Huazhong Agriculture University, Wuhan 430070, China
* Correspondence: ygan@zju.edu.cn; Tel./Fax: +86-517-88982204
† These authors contributed equally to this work.

**Abstract:** Heavy metal toxicity generally refers to the negative impact on the environment, humans, and other living organisms caused by exposure to heavy metals (HMs). Heavy metal poisoning is the accumulation of HMs in the soft tissues of organisms in a toxic amount. HMs bind to certain cells and prevent organs from functioning. Symptoms of HM poisoning can be life-threatening and not only cause irreversible damage to humans and other organisms; but also significantly reduce agricultural yield. Symptoms and physical examination findings associated with HM poisoning vary depending on the metal accumulated. Many HMs, such as zinc, copper, chromium, iron, and manganese, are present at extremely low levels but are essential for the functioning of plants. However, if these metals accumulate in the plants in sufficient concentrations to cause poisoning, serious damage can occur. Rice is consumed around the world as a staple food and incidents of HM pollution often occur in rice-growing areas. In many rice-producing countries, cadmium (Cd), arsenic (As), and lead (Pb) have been recognized as commonly widespread HMs contaminating rice fields worldwide. In addition to mining and irrigation activities, the use of fertilizers and pesticides has also contributed significantly to HM contamination of rice-growing soils around the world. A number of QTLs associated with HM stress signals from various intermediary molecules have been reported to activate various transcription factors (TFs). Some antioxidant enzymes have been studied which contribute to the scavenging of reactive oxygen species, ultimately leading to stress tolerance in rice. Genome engineering and advanced editing techniques have been successfully applied to rice to improve metal tolerance and reduce HM accumulation in grains. In this review article, recent developments and progress in the molecular science for the induction of HM stress tolerance, including reduced metal uptake, compartmentalized transportation, gene-regulated signaling, and reduced accumulation or diversion of HM particles to plant parts other than grains, are discussed in detail, with particular emphasis on rice.

**Keywords:** rice; heavy metals; stress tolerance; molecular breeding





## 1. Introduction

Heavy metal pollution is increasing globally, damaging the environment and posing serious health risks to humans and other living organisms [1]. Heavy metal (HM) toxicity is commonly referred to as pollution caused by some naturally occurring elements known as heavy metals [2]. They have high atomic masses and densities (higher than 4 g/cm$^3$) that are cytotoxic above certain concentrations [3]. Toxic HMs are subdivided into essential HMs and non-essential HMs according to their requirements for normal functioning of life [4]. Arsenic (As), cadmium (Cd), lead (Pb), mercury (Hg), thallium (Tl), and chromium (Cr) are recognized as useless for the normal functioning of life and are classified as non-essential HM. On the other hand, copper (Cu), boron (B), zinc (Zn), iron (Fe), selenium (Se), molybdenum (Mo), and manganese (Mn) are essential for proper life functioning. These are mandatory HMs that are practically required for cell survival [5,6].

The root causes of HM pollution are often attributed to the rapid rise of urbanization, the reckless growth of industrialization, and its waste disposal mechanisms [6,7]. After the industrial revolution and economic globalization, there has been an exponential increase in the diversity of environmental pollutants from various anthropogenic sources [4,8]. Consequently, a series of emerging issues related to food security have become global issues, especially given their close relevance to agriculture and human health. Excessive HMs can affect soil microbiological balance and reduce its fertility [9,10]. All metals, even including essential HMs, inhibit plant growth and metabolism; and reduce yield when certain thresholds are exceeded (Table 1, Figure 1). Plants take up both essential and non-essential HMs through active or passive transport. Essential HMs (along with non-essential HMs) readily penetrate root cells and perform their major functions, but can also be transported to tissue/subcellular sites where they can damage cells and their functions [11]. Once part of the plant's internal machinery, HMs are transported and accumulated in grains, leaves, and other plant parts, where they are consumed by animals, birds, and humans, causing serious health problems. In terms of human exposure, the risks of HM ingestion include permanent brain damage, cirrhosis of the liver, encephalopathy, dementia, hemorrhage, impaired renal function, alveolitis, bronchitis, emphysema, sperm motility, bone disorders, gastrointestinal malignancies, and cardiovascular illness [12,13]. In plants, HM interferes with many metabolic processes such as photosynthesis, water balance, and nutrient absorption, resulting in plant growth retardation, grain yield losses (Table 1), and even death [14].

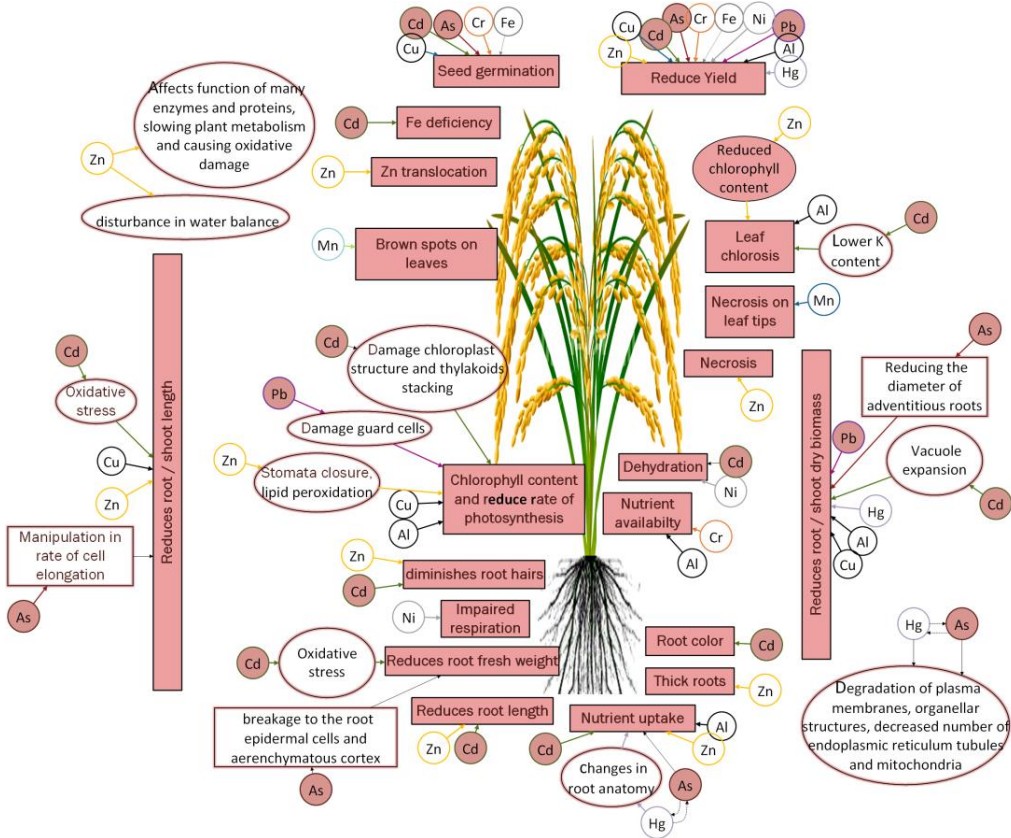

**Figure 1.** Schematic representation of the various negative effects of excess heavy metals on different parts of rice plants.

Rice, an important food source in many parts of Asia, is currently being severely damaged by the repeated use of HM-rich pesticides, excess water, and sludge waste (as organic fertilizer), creating reservoirs of HMs in the topsoil [15,16]. HMs can accumulate in the soil through the leaching processes and dissolve in groundwater, contaminating groundwater

as well as surface water [17]. Several investigative reports indicate that the consumption of rice grown in fields high in Potentially Toxic Elements (PTEs), i.e., contaminants exceeding maximum allowable concentrations, can endanger human health [18,19]. Many researchers have reported that toxic metals such as As, Cd, and Pb along with other essential metals, e.g., Fe, Zn, and Cu are transported from soil and stored in rice grains as well as other plant parts such as the husks, leaves, and stems that are directly or indirectly edible. The accumulation of As and Cd in rice grains has become a serious problem worldwide [20,21]. Under moderately dry soil conditions, excessive Zn and Cd uptake can negatively affect Fe uptake, which also affects photosynthesis and yield [22,23]. Bioaccumulation of HMs in rice leads to biomagnification in the food chain, which can be dangerous as HM causes severe disease in humans [22,23]. Low levels of zinc have been reported to be linked to various physiological functions in the human body, while high levels can potentially offer major health hazards [24]. Likewise, an excess Cu can cause diarrhea, nausea, and Wilson's disease [19]. Therefore, understanding the underlying mechanisms of HM uptake by plants, HM-mediated signaling and distribution in plants, and further exploring various breeding techniques effective in minimizing metal buildup in rice grains is crucial from the perspective of human health. In light of this, this review offers a summary of the most recent research on efforts to reduce HM bioaccumulation in rice grains as well as efficient breeding techniques for developing metal-tolerant rice varieties.

**Table 1.** Rice yield loss due to different HM toxicities reported in different studies.

| Heavy Metal | Dose/Concentration (mg/kg) | Yield Loss (%) | Reference |
|---|---|---|---|
| Pb | 400–1200 | 46–69 | [25] |
| As | 24.5 | 40 | [26] |
| Cd | 50–150 | 34–63 | [27] |
| Fe | 385–1197 | 16–78 | [28] |
| Cu | 100–1000 | 10–90 | [29] |
| Zn | - | 20–40 | [30] |
| Ni | 40–100 | 54–70 | [31] |
| Al | - | 30–60 | [32] |
| Cr | 200–400 | 30–48 | [32] |
| Hg | 4–5 | 50–70 | [33] |

## 2. Natural and Anthropogenic Sources of HM Toxicity in Rice

Depending on the mode of origin, the bioaccumulation of HMs in the environment may be either anthropogenic (caused by humans) or natural. Anthropogenic sources can be subdivided into two main sources, agricultural waste and municipal or industrial waste (Figure 2). Rice grains actively accumulate HMs from contaminated soil. Soil characteristics are, therefore, regarded as the most significant influencing element [34]. Accumulation of HMs in agricultural soils tends to increase with weathering of metal-enriched rocks and other soil-forming processes [35]. HMs can be mobilized beyond the buffering capacity of soils due to saturation or changes in environmental conditions such as land exploitation, agricultural inputs, and climate change [36].

Some chemical fertilizers, such as phosphate fertilizers, can introduce HM into the soil. Trace amounts of HM are also present in chemical fertilizers because they occur as impurities in the raw materials and natural minerals used to manufacture these fertilizers [37]. Phosphogypsum from waste phosphate fertilizers can form a variety of toxic HMs in the soil, which can be transferred to plants. Later these soils become enriched mainly in toxic HMs (e.g., Cd, Cr, and Pb). Excessive application of cadmium-rich phosphorus fertilizers can also be a source of cadmium pollution in rice fields [38]. Consumption of rice from rice fields contaminated with HMs can be a potential source of exposure to toxic HMs, especially cadmium, lead, and arsenic [39]. Consumers may be exposed to health risks and illnesses if they ingest rice that has been farmed in contaminated fields over an extended period of time [40]. A recent study found that high concentrations of Cd, Hg, Pb,

and Zn in soil were positively correlated with As in rice grains, and that As, Cr, Hg, Pb, and Zn increased the Hg content of rice, suggesting that the presence of these elements also promote the accumulation of toxic HMs such as As and Hg in rice [41]. The increasing use of large amounts of organic manure in paddy fields around the world has been identified as a source of HMs in the soil. It was also discovered that the amount of arsenic in rice grains increased when rice was cultivated in soils with high concentrations of As and FYM [42].

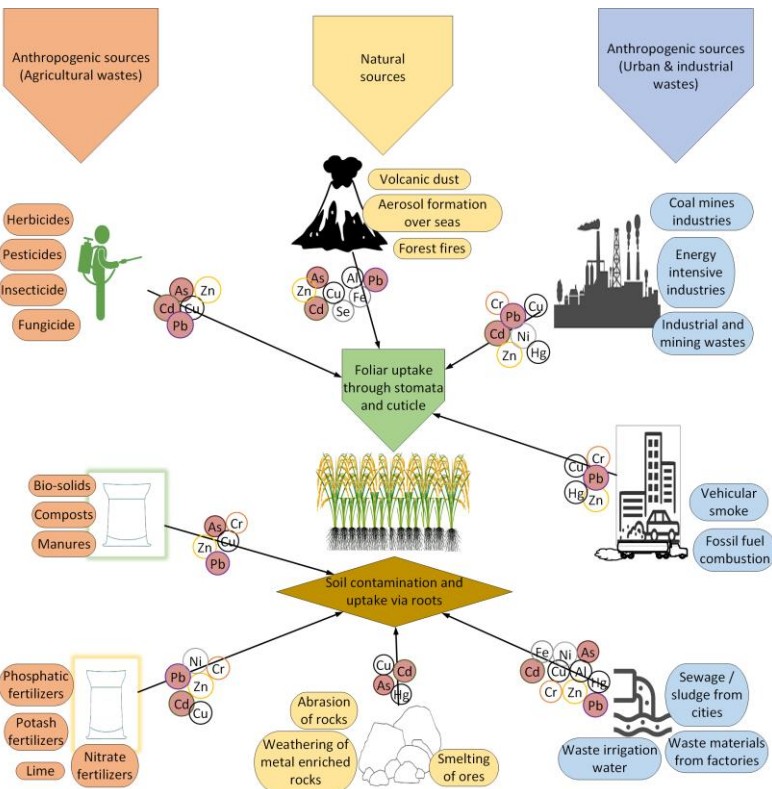

**Figure 2.** Anthropogenic and natural sources of heavy metal pollution in the environment.

To determine the potential sources of HMs and their buildup in agricultural soils, several research investigations have been conducted in different countries. According to a survey report, phosphate fertilizers tested included traces (mg/kg) of 5.6–17.2 Pb, 19.4–273.0 As, 9.5–96.4 Cd, and 0.01–0.42 Hg, and ammonium phosphate, ammonium nitrate, and compound fertilizers tested positive for 50–60 mg/kg of As [43]. Another study found that phosphate fertilizers were a significant source of heavy metals (HMs) entering agricultural soils in England and Wales, particularly zinc, copper, and cadmium. In contrast, soils in Argentina were frequently fertilized with inorganic phosphate fertilizers, which led to the presence of cadmium, chromium, copper, zinc, nickel, and lead [44]. Today, organic fertilizers also contain HMs that find their way into drinking water or are added to bedding (such as straw) in paddy fields around the world. Therefore, the long-term excessive application of organic fertilizers containing HMs such as copper and zinc will also lead to their accumulation in agricultural soils.

An extensive study in eastern China investigated HM levels in urban soils and found that high concentrations of HMs in soil were mainly concentrated in central and western regions with relatively few spatial variations [45]. Municipal solid waste, automobile emissions, combustion of coal, and HM emissions from human activities including vehicle exhaust are all partially responsible for the rise of lead isotopes in urban soils. Whether for adults or children, the risk of HM contamination from soil when ingested is much higher than when it comes in contact with the skin. Another study conducted in Tangchang in 2022 to determine HM contamination (HM) in agricultural soil collected 788 samples of

topsoil and analyzed their HM concentrations [46]. It was further reported that transport and deposition could be sources of Pb and Hg; Zn, Cr, and Ni could come from natural resources and industrial activities, while Cu and Cd came from agricultural activities. These studies additionally indicate that the location of mines is the primary factor influencing the environmental distribution of Zn, Ni, Pb, As, and Cd [3].

### 3. Progress in Molecular Breeding to Develop HMT Tolerance in Rice

The problem of HM toxicity has emerged as one of the global risks to sustainable agricultural output due to the rapid increase in the accumulation of HMs from various sources in agricultural soils during the past few decades. Agricultural experts have expressed interest in further investigating the underlying mechanisms that allow plants to resist HM toxicity [47]. Given a thorough understanding of HM transport and deposition in various plant organelles, researchers have proposed a number of experimental methods using innovative molecular approaches that can assist rice plants develop HM tolerance. In particular, detoxification, transport, and/or sequestration are the primary objectives of HM control techniques [48]. Fluid transport from roots to other plant parts involves water transpiration, root pressure, cation exchange in the cell walls of xylem vessels, formation of complexes with amino acids (Cu), histidine, peptides (Ni), and chelates with organic acids (Zn) [49]. Accordingly, the concentrations of most HMs gradually decreased with distance from the root [37]. These elements are transported inside the plant by cell wall charge interactions and the formation of soluble organic complexes in the sap [50].

### 3.1. Role of HM Transporters (HMTs) and Tolerant Proteins (HMPs)

Previously reported P1B subfamily of HM transporters P-type ATPases (HMAs) affect HM uptake and transport in different plants [51–53]. The researchers found upregulation of HMA genes and retrotransposons in rice after HM treatment and intergenerational inheritance of altered expressions [54]. This study demonstrated how rice plants respond to HM stress by altering locus-specific gene expression and trans-generational inheritance due to altered gene regulation, even after HM stress has been eliminated. Fu et al. linked a G-type ATP-binding cassette transporter *OsABCG36* to Cd tolerance in rice. Cd-induced *OsABCG36* was expressed localized to the plasma membrane in protoplast cells of the root tip and mature roots. Knockdown of *OsABCG36* increased Cd accumulation in roots and enhanced sensitivity to Cd stress without affecting tolerance to other HMs (i.e., Al, Zn, Cu, and Pb) [55].

Metal tolerance proteins (MTPs), another type of membrane protein, have also been demonstrated to be involved in metal transport and to impart tolerance to certain HMs in plants. In a comprehensive molecular study of rice MTP genes, three thousand rice genotypes were investigated using available genome-wide resequencing information, highlighting the evolution and allelic diversity of MTP genes [56]. These findings indicated that *MTP1*, *MTP8.1* and *MTP9*, *MTP11* localize to QTL/m-QTL for Zn and Cd accumulation, respectively. *MTP9* and *MTP8.1* had specific root and shoot expression, respectively, at all stages of the rice plants. After getting exposed to toxic HMs, it was determined that some seed-specific MTPs regulated metal transport during the seed-filling stage downregulated the expression of the majority of MTP genes in the roots, and upregulated those in the shoots, indicating that MTP employs various mechanisms in different tissues.

Certain HM-associated proteins (HMPs) have been reported to be involved in HM detoxification. In earlier studies, the novel genes were found to regulate a protein known as cell number regulatory factor (CNR) in plants e.g., *TaCNR2* in wheat and *ZmCNR1* in maize [57]. This protein, like the cadmium-tolerant protein in plants, modulates cell number and HM transport. Overexpression of *TaCNR2* in rice also increases stress resistance to Cd, Zn, and Mn, and their translocation from roots to shoots. *TaCNR2* has been shown through experimental studies to transport HM ions, providing another source of genes that improve nutrient absorption and decrease the buildup of hazardous metals in rice plants [58]. Li et al. also identified 46 HMPs in rice, which they named *OsHMPs* 1–46 according to their

chromosomal locations. Studies have shown that HMPs are governed by different TFs, and only 8 *OsHMPs* are assembled in rice tissues. Among them, *OsHMP37* had a much higher expression level than the other seven HMPs, whereas *OsHMP28* was solely expressed in the roots. Only *OsHMP09*, *OsHMP18*, and *OsHMP22* showed increased expression levels in all tissues, despite the fact that most of the selected *OsHMPs* were variably expressed in different tissues under varying HM exposures [45]. Approximately 43 putative Fe–S cluster assembly genes have also been identified in the rice (*Oryza sativa*) genome and the expression of all genes has been validated [59,60]. Liang et al. investigated the role of Fe–S cluster assembly in leaf chloroplasts are particularly sensitive to HM treatment, and genes encoding Fe–S cluster assembly in roots are sensitive to iron toxicity, oxidative stress, and HM stress; and are shown to be upregulated in response [60]. *OsHMA2* is a major Zn/Cd transporter identified in rice roots and has been experimentally shown to promote Zn/Cd transfer from roots to shoots. [61].

### 3.2. Role of microRNAs (miRNAs)

In several investigations, high-throughput sequencing and miRNA microarray data have indicated the involvement of several miRNAs in the responses of various plant species to toxic HMs [62]. Overexpression of miRNAs or the development of miRNA-resistant target genes have been widely utilized to demonstrate the role of metal-responsive miRNAs, and these miRNAs and their target genes are an essential component of a large-scale regulatory network controlling different metabolic processes in response to metal stress [63,64]. Several regulatory networks involving miRNAs and their TFs have been studied [35]. Ding et al., identified twelve (12) Cd-responsive miRNAs in rice using miRNA microarray assays. Expression of another Cd-responsive miRNA (miR166) was significantly suppressed under Cd exposure at the seedling stage and its overexpression reduced Cd-induced oxidative stress [65]. Overexpression of miR166 also reduced root-to-shoot Cd translocation and Cd accumulation in rice grains [61,66].

### 3.3. QTL and Fine Gene Mapping

Researchers have extensively used genetic markers in rice crops to identify genomic loci involved in various plant physiological functions under simulated stress conditions and associated tolerance mechanisms [67,68]. Breeders have used the same approach to identify QTLs associated with specific HM accumulation, translocation, and tolerance strategies in rice to reduce HM toxicity [69]. Arsenic is the most toxic of all HM and rice absorbs more arsenic than other crops [70]. Once the crop has absorbed HM through its roots, it is distributed throughout the plant and then transferred to the rice grain [71]. The International Agency for Research on Cancer (IARC) declared As, Cd, and Cr to be highly carcinogenic and can damage DNA by disrupting DNA synthesis/repair mechanisms and resulting in neuropsychiatric disorders [72]. Due to climate change, more arsenic is expected to be present in the environment as more rain releases arsenic and other HM previously trapped in mining areas [73]. Ingestion of 1 mg of arsenic per day can affect the skin for several years [74]. Therefore, As has now become the most priority research area among plant breeders. Numerous QTL mapping and genome-wide association studies (GWAS) have documented a number of QTLs/candidate genes that confer toxicity tolerance and HM accumulation in rice (*Oryza sativa* L.) [3]. In an investigation, an early backcross population from a cross between WTR1 (indica) and Haoannon (japonica) was grown hydroponically and exposed to As (10 ppm) for seven days and genotyped using 704 SNP markers. One QTL for relative chlorophyll content was identified on chromosome 1, two QTLs for arsenic content in roots were discovered on chromosome 8, and six QTLs for arsenic content in shoots were identified on chromosomes 2, 5, 6, and 9 [75].

Tyagi et al. [76] found that the tolerant genotypes accumulated less Al and showed a minimal reduction in root-shoot biomass under Al stress compared to susceptible genotypes. Transcriptomic data indicated that tolerant genotypes conserved energy by down-regulating key glycolytic pathway genes in roots, maintaining transcription levels of key

energy-releasing enzymes, up-regulated signaling, and regulatory transcripts encoding zinc finger proteins and cell wall-associated transcripts under Al stress. Stein et al. [77] also designed a similar study to identify candidate genes for iron toxicity tolerance by investigating the effect of excess iron on two rice cultivars with apparent tolerance to iron toxicity. These results suggest that physiological and anatomical changes and HM permeability in various parts of rice may be related to toxic tolerance.

Shilin et al. [78] also investigated the genetic mechanism of Cd tolerance in rice, using a RIL population derived from crossing two parents (i.e., PA64s and 93–11) to analyze the associated QTLs to cadmium tolerance at the seedling stage. Two QTLs, i.e., *qCDSL1.1* and *qCDSL1.2* were identified in Hangzhou and Lingshui, respectively. The Chromosomal Segment Replacement Line, *CSSLqCDSL1* was developed in the context of parent 93–11 which contained *qCDSL1.1/qCDSL1* from parent PA64. Under Cd stress conditions, *CSSLqCDSL1* had a longer shoot length compared to 93–11, demonstrating tolerance to Cd toxicity. In another investigation by Maghrebi et al. [79] two major rice cultivars (i.e., Capataz and Beirao) with different Cd tolerance were exposed to various Cd concentrations (0.01, 0.1, and 1 μM) and their potential capacities to sequester and transfer Cd. QTL mapping for toxic metal/metalloid stress tolerance in rice identified three QTLs associated with As, one associated with Cu and Hg, and two associated with Fe and Zn content.

In a genome-wide association study (GWAS), 188 different cultivated rice germplasms were examined at the seedling stage under normal and Cd stress conditions. By combining GWAS data, transcriptome analysis, database gene annotation intelligent Gene Ontology (GO), and homologous gene annotation and function, 148 candidate Cd-mediated growth response (CGR) genes were obtained [21]. In another GWA study, the correlation between arsenic and eight essential ions in the rice germplasm population was analyzed and it was shown that the association between arsenic and other essential ions was affected by growing conditions and various genetic factors. A cis-eQTL for *AIR2* (arsenic-induced RING finger protein) was isolated by transcriptome-wide association studies, and the expression level of *AIR2* was confirmed to be lower in indica than in japonica. By genome-wide analysis, arsenic-associated QTLs were discovered on chromosomes 5 and 6 under submerged and intermittent submerged conditions [70].

Another GWA study on low Cd accumulation in rice identified the *OsABCB24* gene as the basis of a new QTL (*qCd1-3*) [80]. GWAS of trace iron and zinc in rice grains revealed novel associations of marker traits with 2.1–53.0% phenotypic variation, which may help identify candidate genes for increased iron tolerance and zinc [81]. Fe toxicity in rice crops was also investigated using a GWAS approach, and three linkage disequilibrium (LD) blocks were found to contribute primarily to Fe omission on chromosomes 1, 2, 3, 4, and 7 [82]. Other GWA studies found 22, 17, and 21 QTLs in rice associated with As, Cd, and Pb toxicity, respectively. To identify genomic areas regulating the covariance between mineral elements in the rice genome, Liu et al. used multivariate QTL analysis and principal component analysis (PCA). They sequenced the entire genome of rice RILs and identified potential candidate loci under the QTL clusters, including *OsHMA4* and *OsNRAMP5* [83].

### 3.4. HM Tolerant Transgenic Rice

Over the past decade, genetically modified (GM) crops have been officially adopted in many countries around the world, with their coverage increasing rapidly every year. Therefore, extensive research has been conducted on genetically modified (GM) rice in particular, focusing on the development of rice species tolerant to various biotic and abiotic factors [84]. Researchers are now also focusing on developing genetically modified rice by inserting new genes, which could reduce the buildup of HM in rice grains [85,86]. For instance, glutathione peroxidase (*PgGPx*) was discovered to impart salt and drought resistance to transgenic rice plants in a previous investigation [85]. The peroxidase gene family plays an essential role in maintaining ion homeostasis because it transports metal ions, including Cd [87,88]. Elaborated the role of *PgGPx* on Cd stress in rice and found that *PgGPx* transcript level was strongly up-regulated in response to exogenous Cd level. Under

Cd stress, overexpression of *PgGPx* in transgenic rice improves control of ROS scavenging pathways and cellular ion homeostasis maintenance.

Another multigene family of plant-specific peroxidases (Class III) is involved in various physiological and developmental processes and tolerates abiotic stresses and HM such as aluminum, zinc, cadmium, and copper by removing ROS and RNS [89]. In a study, Kidwai et al. [90] identified a class III peroxidase (*OsPRX38*) from rice that was consistently increased in response to arsenate (As) and arsenite stress. Overexpression of *OsPRX38* in transgenic rice significantly increased arsenic (As) tolerance by regulating lignin biosynthesis which acts as an apoplast barrier for As entry into root cells, resulting in reduced As accumulation in transgenics. Another protein (*MTH1745*) from the thermophilic archaea *Methanothermobacter thermoautotrophicum* has already been shown to prevent citrate synthase aggregation after heat denaturation and to play an important role in heat stress resistance [35]. Overexpression of *MTH1745* in transgenic rice plants also increased mercury tolerance and increased photosynthesis rate compared to non-transgenic rice. Chen et al. [91] & Ding et al. [92] down-regulated the expression of miR5144-3p and overexpressed *OsPDIL1-1*, respectively, in transgenic rice plants which significantly improved mercury stress tolerance.

Researchers improved rice varieties low in As, Cd, and Cs by eliminating the genes (such as *OsNRAMP5*, *OsNRAMP1*, *OsARM1*, and *OsHAK1*) [93,94]. Other transporters such as *OsNRAMP1* have been identified as a mediator of Cd and Mn uptake [95]. Overexpression of another gene, *LmMTP1*, metal resistance protein 1 (MTP1), isolated from ryegrass, resulted in increased resistance to Zn, Co, and Cd in transgenic lines [1]. Overexpression of microRNA (miR166) enhanced tolerance to Cd toxicity in transgenic rice plants thereby reducing Cd accumulation in grains [65]. Plants overexpressing *OsHB4* have been shown to have increased sensitivity to cadmium and accumulate cadmium in their leaves. Conversely, *OsHB4* silencing increased the tolerance of transgenic plants to Cd.

In a study, to limit arsenic in the roots of rice plants as a detoxification mechanism, the phytochelatin synthase *CdPCS1* from cornflower, an arsenic-accumulating aquatic plant, was transgenically expressed [86]. Increased arsenic accumulation in roots and shoots was observed in transgenic rice lines. However, compared to non-transgenic plants, all transgenic lines accumulated much less arsenic in grain and hulls. Huang et al. [96] identified *OsHMA4* in yeast as a possible causative gene of a QTL controlling Cu accumulation in rice grains and provided evidence that *OsHMA4* functions as a Cu chelate in root vacuoles, limiting the accumulation of Cu in grains. The resulting transgenic lines were more resistant to HM stress than the nontransgenic or wild type.

Gu et al. isolated *DEFENSIN 8 (DEF8)*, a defensin-like gene expressed in rice grains, and suggested that *DEF8* promotes Cd loading into the xylem and mediates Cd translocation from root to shoot and subsequent distribution in the grain [97]. *DEF8* is also a modulator of Cd unloading to the phloem without affecting the accumulation of essential mineral nutrients and key crop traits [98]. Li [45] reported that low cadmium accumulation rice lines could be constructed by overexpressing a truncated *OsO3L2* gene. Researchers have successfully developed rice plants that reduce cadmium accumulation by overexpressing the rice genes *OsO3L2*, *OsO3L3*, or their truncated versions [62]. Since the tissue localization of both these genes showed high expression in roots, it seems possible that roots are involved in low Cd accumulation. According to another study, transgenic rice plants overexpressing V-PPase accumulated more cadmium in the roots than in the shoots [99]. Similarly, another mutant line overexpressing the *OsHIPP16* (OE) gene significantly improved rice growth under Cd toxicity stress [100].

*3.5. CRISPR/Cas Genome Editing*

In the past decade, advanced genome engineering technology, CRISPR-Cas9 has also been used to improve detoxification efficiency in rice by targeting HM-specific genes [101, 102] (Table 2). Gene function and the way in which it affects other biochemical processes are changed by indel mutations and targeted substitutions generated through the CRISPR/Cas9

system. The *OsNRAMP1* gene that controls the uptake of various HMs (including Cd, Fe, As, and Mn) in various crops was successfully deleted by the gene editing approach, which also greatly reduced the uptake and storage of Cd and As in the grains of rice [103]. Cd accumulation in rice was studied by deleting a segment of the *OsABCG36* gene by CRISPR/Cas9 technology. Cd tolerance developed in knockout mutants as a result of Cd accumulation in root cells and excretion of the Cd content from the cytosol to detoxify its effects [55]. The *OsLsi* gene family has been reported to be involved in Si xylem loading in rice roots, which is required for efficient Si transport from roots to stems. CRISPR-Cas-based inactivation of this gene (*OsLsi*) resulted in reduced Si uptake in xylem sap and also reduced As accumulation in rice [33]. Similarly, CRISPR/Cas9 has been used to knock down *OsNramp5* and *OsLCT1* to reduce cadmium (Cd) accumulation [104] and inactivate the low cesium (Cs) K+ transporter *OsHAK1* in rice plants [94]. Overexpression of gene *OsLCT2* (a low-affinity cation transporter) using CRISPR-Cas9 also reduced Cd accumulating in rice grains [105]. More recently other genome-editing technologies, such as CRISPR-Cas12a, CRISPR-directed evolution, and base-editors, have also succeeded in more efficiently multiplex genome-editing in rice [60].

**Table 2.** List of targeted genes modified for HM tolerance in rice using CRISPR-Cas9 genome editing technology.

| Metalloid | Targeted Gene(s) | Molecular Functions | Role in HM Toxicity Tolerance | References |
|---|---|---|---|---|
| As, Si | *OsLsi* | Silicon (Si) transporter | Low arsenic uptake | [33] |
| Cd | *OsABCG36* | G-type ATP-binding cassette (ABC) transporter | Cadmium sequestration and toxicity tolerance | [55] |
| Cs | *OsHAK1* | High-Affinity Potassium (K+) Transporter | Low cesium accumulation in root and shoots | [94] |
| As, Cd | *OsNRAMP1* | Cadmium, Iron, and manganese uptake/transporter | Low arsenic and cadmium content in grains | [103] |
| Cd, Pb, Mn, Fe | OsN*RAMP5* | Major transporter for metal uptake | Low cadmium content in grains | [104] |
| Cd | *OsLCT1* | Low-Affinity Cation Transporter | Low cadmium uptake | [104] |
| Cd | *OsLCT2* | Low-Affinity Cation Transporter | Low cadmium accumulation in grains | [105] |

## 4. Breeding Tools for Improving HM Toxicity Tolerance in Rice

### 4.1. Physiological Screening

The study of natural variation in related traits is of great importance for the breeding and development of new crop varieties. Previous studies have identified wide genetic diversity for various HMs (e.g., Zn, Cu, Cd, Pb, Ni, As, and Sn) based on physiological criteria (e.g., relative shoot weight, root weight, and chlorophyll) [106]. One of the most effective screening strategies for developing HM-tolerant rice genotypes is to test for HM tolerance using valid screening systems (e.g., a hydroponic system) to estimate the tolerance level. The copper tolerance of 16 hydroponic rice cultivars was studied and four rice genotypes (Xiushui 123, Xiushui 134, Jiahe 991, and Xianghu 301) showed improvements in root length, root dehydrogenase activity, and root dry weight under copper stress, and the screened genotypes were used for local cultivation [107]. Another study found that under arsenic stress, the rice variety BRRI-29 from the Bangladesh Rice Research Institute (BRRI) had the best germination rate, biomass, and longest roots [108]. Genotypes with higher proline content, leaf chlorophyll content, and flag leaf area are also considered effective physiological screening indicators for lead tolerance in rice. Natural rice populations grown

around mine sites have been reported to be more tolerant to HM [109], and their use in breeding programs to increase HM tolerance is being studied.

### 4.2. Mutation Breeding

One of the main approaches to understanding plant responses to HM is to use mutants with altered sensitivity or tolerance to HMs. Mutation breeding is an important tool for generating genetic diversity for desired traits, including the development of new varieties with increased tolerance to HM [110]. In several mutagenesis studies, functionally deficient mutants of some HM tolerance genes were successfully obtained, which significantly reduced the accumulation of the associated HM in different parts of the rice plants [111]. Common methods of mutagenesis include EMS mutagenesis, space mutagenesis, and irradiation mutagenesis [112]. The expression of these genes has been silenced or modified by knocking out their alleles or promoters to study their effect on HM accumulation or translocation efficiency in plants, which can be used to develop genotypes with reduced HM accumulation, making them tolerant to HM toxicity. Several transporter genes have also been identified to mediate HM uptake, transport, or distribution in rice plants [93,95,113].

### 4.3. Molecular Gene Mapping

Different molecular studies such as genome-wide association studies (GWAS) have been widely used as effective tools to exploit elite alleles controlling important agronomic traits in rice germplasm [38,39] (Table 3). Researchers have also used GWAS in several molecular screening studies to detect a large number of QTLs for HM accumulation in rice grains collected from rice germplasm [41,42]. Several molecular studies have investigated the response of rice plants to high HM stress [114]. Based on GWAS and transcriptome analysis, tolerance germplasm resources were investigated, and superior natural variants associated with HM resistance were identified [113,115]. Collectively, the identified QTLs serve as a resource to potentially improve tolerance to HM toxicity in today's major cultivars and provide useful information for the use of these QTLs/genes in future breeding programs.

**Table 3.** Identified genes/QTLs associated with heavy metal tolerance in rice.

| SR | Genes/QTL | Metalloids | Mechanism Involved | Ref |
|----|-----------|-----------|--------------------|-----|
| 1 | *OsNRAMP5* | Fe and Mn, as well as Cd and Zn | Overexpression of *OsNRAMP5* to increase tolerance to Cd toxicity | [93] |
| 2 | *OsHMA3* | Cd | Overexpression of *OsHMA3* to increase tolerance to Cd toxicity | [116] |
| 3 | *OsABCG31* | Cd and Pb | Overexpression of *OsABCG31* enhanced their tolerance to Cd and Pd toxicity | [117] |
| 4 | *OsLCT1* | Al | Overexpression of *OsLCT1* to increase tolerance to Al toxicity | [106] |
| 5 | *OsSIZ* | Cd | Overexpression of *OsSIZ1* enhanced tolerance to Cd toxicity. | [107] |
| 6 | *OsZIP1* | Zn | Overexpression of *OsZIP1* can enhance tolerance to Zn toxicity | [108] |
| 7 | *OsNAC5* | Cd and Pb | Overexpression of *OsNAC5* enhanced their tolerance to Cd and Pb toxicity | [109] |
| 8 | *OsMT1e* | Cd and Zn | *OsMT1e* encodes a metallothionein protein involved in metal detoxification | [118] |
| 9 | *OsNAC2* | Cd | *OsNAC2* is a transcription factor that is involved in regulating the expression of genes involved in stress responses in rice | [111] |
| 10 | *qNRG2-1* | Ni | *qNRG2-1*, a QTL on chromosome 2 is associated with Ni tolerance in rice associated with the expression of the *OsNramp5* gene | [119] |
| 11 | *qCdt1* | Cd | *qCdt1-2*, a QTL on chromosome 1 is associated with Cd tolerance with overexpression of *OsHMA3* | [110] |
| 12 | RM219 | As | accumulation of As | [112] |
| 13 | *qPC1* | Cd | *qPC1* QTL was associated with lower Cd accumulation in rice grains | [113] |
| 14 | *OsIRO2* | Cd | *OsIRO2* is a transcription factor regulating the expression of genes involved in Fe homeostasis in rice | [120] |

**Table 3.** *Cont.*

| SR | Genes/QTL | Metalloids | Mechanism Involved | Ref |
|----|-----------|-----------|--------------------|-----|
| 15 | *qGCD7* | Cd | This QTL has been found to be important for rice tolerance to Cd toxicity | [121] |
| 16 | *qHLR1* | Cd and Pb | This QTL has been found to be important for rice tolerance to both Cd and Pb toxicity | [122] |
| 17 | *qCST11* | Pb | This QTL is associated with Cd and Pb tolerance in rice identified through a GWAS | [80] |
| 18 | *OsIRT1* | Fe | This gene encodes an Fe transporter involved in Cd uptake in rice. It was found to be upregulated under Cd stress | [123] |
| 19 | *qMRS6.1* | Mn | This QTL is associated with manganese (Mn) tolerance in rice. It was identified through a GWAS | [114] |
| 20 | *OsPCS1* | Cd | This gene encodes phytochelatin synthase involved in the synthesis of phytochelatins, a group of peptides that bind to HMs and detoxify them in plants. It was found to be upregulated under Cd stress in rice. | [115] |
| 21 | *qHTSF4.1* | As | The QTL is associated with Cd and As tolerance in rice | [116] |

*4.4. Gene Silencing*

Several investigation reports have been published regarding the development of new rice lines with low HM accumulation in grains by knocking out the HM transporter genes using different gene silencing techniques such as RNA interference (RNAi) [124], CRISPR/Cas9 system [93,125] and transgenics. Li et al. [124] successfully reduced cadmium (Cd) accumulation in rice seeds by inhibiting the expression of the phytochelatin synthase (PCS) gene *OsPCS1* using RNA interference (RNAi). Two mutant lines (pmei12-J3 and pmei12-J8) were obtained by editing the *OsPMEI12* gene using CRISPR/Cas9 editing technology, and the cadmium stress resistance of the pmei12 line was found to be significantly improved [30]. Similarly, silencing of genes such as *OsNramp5* and *OsLCT1* has been reported to reduce Cd accumulation [104] and *OsHAK1* (low Cs K+ transporter) [94] to attenuate Cs accumulation in rice. Related studies on genome editing via CRISPR-Cas technology are shown in Table 2 above.

**5. Challenges and Limitations in Breeding for HM-Tolerant Rice**

Compared to other abiotic stresses, HM toxicity tolerance is genetically more complex and involves many components of signal transduction and translocation pathways that are polygenic in nature. Therefore, it is relatively more difficult to control and reproduce [126]. Moreover, genetic manipulation of HM-responsive genes, metabolites, and proteins has yielded unexpected results, but its potential has not been fully exploited due to complex genetics and insufficient understanding of their epigenetic behavior [127]. Several other factors, such as environmental influences and the significant interrelationships among various HMs, have been reported to negatively affect the outcome of effective breeding for the development of HM-tolerant plants [128]. Several metal-uptake and transporter genes are associated with more than one HM. For example, *OsNRAMP1* and *OsNRAMP5* are involved in the uptake/transport of cadmium, iron, and manganese [95], and gene editing of HM tolerance targeting a specific HM can also lead to nutritional deficiencies due to insufficient absorption of other essential metals. To counter this, some weak alleles of HM transporter genes (such as *OsNRAMP5*) have also been identified that confer low Cd uptake without causing Mn deficiency in rice [105]. Field screening of HM for tolerance to specific HMs is also very difficult due to other factors such as the large breeding population size, associated field management, and labor costs [129]. In recent decades, considerable work has been done on the detoxification of HMs in soils through phytoremediation, chemicals, bio-chemicals, and microorganisms [124]. However, these strategies are not cost-effective due to the large area on which rice and other crops are grown. Although many measures have been taken to reduce the release of HMs into the environment, the accumulation of HMs in the environment continues to increase rapidly [130].

## 6. Conclusions and Future Prospects

As the population grew, the demand for rice cultivation, which required fertile land and large amounts of water, increased dramatically. Unfortunately, a lot of toxic metals such as, arsenic, lead, and cadmium, some of which are carcinogenic pollutants with a long half-life and are not biodegradable, are contaminating rice fields. Phytoremediation (also known as detoxification of polluted environments) has been widely studied by different researchers to show its potential against HM stress. However, developing natural plant tolerance to HM is more sustainable, less expensive, and more environmentally friendly. Therefore, there is an urgent need to search for new genes to develop rice varieties that are more tolerant to HM toxicity to avoid drastic yield decline due to HM accumulation in the soil and fatal diseases caused by exposure to humans and other organisms.

This review summarizes recent developments and advances in molecular science to increase our understanding of the genetic mechanisms of adaptive resistance in the context of HM toxicity and to accelerate the development of rice genotypes more tolerant to HM toxicity. To date, tolerance to some HM has been shown to be moderate and insufficient to fully mitigate the future impact of HM on food security and human health. Additionally, efforts have been made to increase the tolerance of rice to HM stress to reduce the uptake and accumulation of HM in the grains. However, the complete identification of the master transcriptional regulators that activate the entire signal transduction cascade is still pending. Using molecular approaches to select climate-tolerant plants with excess metal accumulation may open up new possibilities to combat HM toxicity.

In recent years, various advanced technologies and molecular breeding tools have been widely used to better understand the absorption, transport, and tolerance mechanisms of HMs in rice, in order to breed new rice varieties with better tolerance to HM toxicity. Recently, more advanced genome engineering technologies such as CRISPR-Cas9 CRISPR-Cas12, Base Editing (BE), etc. have also been used to improve the efficiency of HM detoxification by modifying various biochemical pathways. Similar to this, other transcriptome approaches have been employed to examine and quantify the abundance of important transcripts in cells from various rice parts. Studies have demonstrated that changes in local gene expression caused by stress in various tissues can result in the generation of novel stress proteins, stress-mediated metabolic compounds, or genes encoding TFs, enhancing the efficiency of plants in regulating HM-stress responses.

An emerging area of research with great potential for commercial application is the creation of genetically modified plants with an excellent ability to chelate certain metals and prevent their deleterious effects. Further research is needed to find specific transporters for specific HMs to facilitate the development of rice varieties tolerant to specific HMs. In the future, bioremediation of contaminated sites could benefit from the coordinated use of traditional breeding techniques combined with molecular biology. Additionally, the use of genome sequencing to identify genes associated with metal tolerance may pave the way for the construction of transgenes with desired properties that can be used in plant extraction techniques. These findings, coupled with in-depth evolutionary studies of putative genes associated with HM tolerance, could yield encouraging results.

**Author Contributions:** Conceptualization, Y.G.; Investigation, Z.H., I.A. and S.Z.; Resources, Y.G., Z.H. and I.A.; Data Curation, Z.H. and I.A.; Writing—Original Draft Preparation, Z.H., I.A. and S.Z.; Writing—Review & Editing, Y.G., Z.H. and I.A.; Supervision, Y.G.; Funding Acquisition, Y.G. All authors have read and agreed to the published version of the manuscript.

**Funding:** The research was funded by Zhejiang Provincial Natural Science Foundation of China (Grant No. LZ22C130002); National Key R & D Program of China (2021YFF1000400; 2022YFD1200403); and National Natural Science Foundation of China (Grant No. U2202204).

**Institutional Review Board Statement:** Not applicable.

**Data Availability Statement:** Data sharing is not applicable to this article.

**Acknowledgments:** The authors acknowledge Zhejiang University and funding sources for the financial support in publishing this paper.

**Conflicts of Interest:** The authors declare no conflict of interest.

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
