# Peer review of "Recent Developments in Rice Molecular Breeding for Tolerance to Heavy Metal Toxicity"

_agriculture, doi:10.3390/agriculture13050944_

Round 1
Reviewer 1 Report
The MS "Recent Developments in Rice Molecular Breeding for Tolerance to Heavy Metal Toxicity" significantly contributes to the field. The strength of the paper is that it is nicely summarizes the current progress in Heavy Metal toxicity tolerance in rice. However, a vast of work already published on the same topic then what is the novelty of this work please justify.
The manuscript can be improved by adding more detailed information regarding methodology and tools of screening and mapping the heavy metal toxicity tolerant genotypes apart from genomics and QTLs.
Challenges and Limitation of developing cultivars against HM toxicity tolerance should be more deeply discussed
1. Please add status data of yield and productivity loss/reduction by heavy metal toxicity globally.
2. Major challenges in developing HM tolerance crops.
Author Response
Response to Reviewers
Dear Editor:
Thank you very much for editing our manuscript “Recent Developments in Rice Molecular Breeding for Tolerance to Heavy Metal Toxicity”. We greatly appreciate the expert comments and suggestions from the reviewers, which help us to improve our manuscript. We have considered the reviewers’ comments based on our review manuscript. The following paragraphs contain our responses to the comments and outlined the changes to the manuscript which have been made accordingly.
Reviewer Comments:
We greatly appreciate the expert comments and suggestions from the reviewers, which help us to improve our manuscript. We have considered the reviewers’ comments based on our review manuscript. The following paragraphs contain our responses to the comments and outlined the changes to the manuscript which have been made accordingly.
Reviewer: 1
The MS "Recent Developments in Rice Molecular Breeding for Tolerance to Heavy Metal Toxicity" significantly contributes to the field. The strength of the paper is that it is nicely summarizes the current progress in Heavy Metal toxicity tolerance in rice. However, a vast of work already published on the same topic then what is the novelty of this work please justify.
Answer: Different reviews on heavy metal tolerance strategies (i.e. agronomy, soil phytoremediation, nanoparticles, etc.) have been published on various crops; however, the data published is outdated. Morover, breeding for HM tolerant rice is a vast topic and new technologies to develop HM-tolerant plants are rapidly being introduced. Previously published articles have not justifiably covered the topic on rice crop breeding for HMT tolerance in rice. In this review article, we include all current literature related to recent developments and advances in all related fields of breeding and molecular sciences used for the induction of HM stress tolerance in rice, including reduction of metal uptake, compartmental transport, regulatory gene signaling and reduction of HM grain accumulation, with particular emphasis on rice crop.
Major questions:
- The manuscript can be improved by adding more detailed information regarding methodology and tools of screening and mapping the heavy metal toxicity tolerant genotypes apart from genomics and QTLs?
Answer: We appreciated the expert comments and suggestions. We have added detailed information under heading “Breeding tools for improving HM toxicity tolerance in rice” including subheadings “Physiological screening”, “Mutation breeding”, “Molecular screening”, “Gene silencing”, covering all the details regarding advance methodology and tools of screening and mapping the heavy metal toxicity tolerant genotypes.
- Challenges and Limitation of developing cultivars against HM toxicity tolerance should be more deeply discussed?
Answer: We have added as new heading “Challenges and limitations”, including all the details regarding the challenges and limitations of breeding HM tolerant rice cultivars.
- Please add status data of yield and productivity loss/reduction by heavy metal toxicity globally.?
Answer: Table is added as table 1: Rice yield loss due to different HM toxicities reported in different studies.
- Major challenges in developing HM tolerance crops?
Answer: Added brief information regarding rice crop and associated major challenges in developing HM tolerance rice genotypes.

Reviewer 2 Report
The authors presented a very a hot topic. All the presented materials are up to the scientific standards but I have the following concerns on this:
1. How can rice molecular breeding for heavy metal tolerance contribute to sustainable agriculture and environmental protection?
2. What are the potential environmental and socioeconomic benefits of developing heavy metal-tolerant rice cultivars?
3. What are the major challenges and limitations facing rice molecular breeding for heavy metal tolerance, and how can they be addressed?
4. Which specific genes and genetic markers have been identified as important for rice tolerance to heavy metal toxicity? Please add a table related to this.
Minor Concerns:
Authors mentioned "A number of QTLs associated 28 with HM stress signals from various intermediary molecules have been reported to activate various transcription factors." but not enlisted any single one.
The color scheme in the legends of Figure 1 is not clear. Please make it more clear.
"A study conducted in 2020 at eastern China examined HM level in urban soil. Considering the environmental and health risks" please provide a reference.
in line 321: OsABCB24 gene it must be italic. please check the whole manuscript carefully.
Add a separate conclusion section with solid recommendations.
Author Response
Response to Reviewers
Dear Editor:
Thank you very much for editing our manuscript “Recent Developments in Rice Molecular Breeding for Tolerance to Heavy Metal Toxicity”. We greatly appreciate the expert comments and suggestions from the reviewers, which help us to improve our manuscript. We have considered the reviewers’ comments based on our review manuscript. The following paragraphs contain our responses to the comments and outlined the changes to the manuscript which have been made accordingly.
Reviewer Comments:
We greatly appreciate the expert comments and suggestions from the reviewers, which help us to improve our manuscript. We have considered the reviewers’ comments based on our review manuscript. The following paragraphs contain our responses to the comments and outlined the changes to the manuscript which have been made accordingly.
Reviewer: 2
The authors presented a very a hot topic. All the presented materials are up to the scientific standards but I have the following concerns on this:
- How can rice molecular breeding for heavy metal tolerance contribute to sustainable agriculture and environmental protection?
Answer: Arsenic, cadmium, lead, and mercury are four heavy metals with no biological role in humans. However, these metals are commonly used in industrial applications and consumer products including pesticides, fundicides and fertilizers being extiesively used in agriculture world-wide. Since these elements are not biodegradeable, their potential toxic effects may be long-lasting within the environment. These heavy metals have the potential to accumulate in vital organs such as the brain, heart, and kidney where they may disrupt normal cellular functioning and if exposures are repetitive or of high concentration, toxicity may result. HM pollution is increasing in an alarming rate as reported by different organisations. All metals, even including essential HMs, inhibit plant growth and metabolism; and reduce rice yield when certain thresholds are exceeded. Once part of the plant's internal machinery, HMs are transported and accumulated in grains, leaves, and other plant parts, where they are consumed by animals, birds, and humans, causing serious health problems. In terms of human exposure, the risks of HM ingestion include permanent brain damage, cirrhosis of the liver, encephalopathy, dementia, hemorrhage, impaired renal function, alveolitis, bronchitis, emphysema, sperm motility, bone disorders, gastrointestinal malignancies and cardiovascular illness. Development of HM tolerant rice would help mitigate the problems associated with HM toxicity in rice grains and improve yield where HM toxicity is a problem limiting rice yield.
- What are the potential environmental and socioeconomic benefits of developing heavy metal-tolerant rice cultivars??
Answer: HM toxicity is increasing as the expansion of industries and use of HM containing pesticides have increased. Reducing human illness caused by heavy metal toxicity in many countries with high levels of toxic HM paddy soils where rice is used as staple. Reduction is yield losses in rice crop due to HM toxicity in soils where rice is being grown or can be grown. There are a number of research in different countries including China, Australia, India, Bangladesh, USA and many other Asia and European countries which found larger areas with paddy soils contaminated with HM pollution above the threshold limits and rice grown in such soils is un-healthy and may cause illnesses to consumers.
- What are the major challenges and limitations facing rice molecular breeding for heavy metal tolerance, and how can they be addressed?
Answer: Added with brief information regarding rice crop and associated major challenges in developing HM tolerance rice genotypes as new heading “Challenges and limitations in breeding for HM tolerance rice”.
- Which specific genes and genetic markers have been identified as important for rice tolerance to heavy metal toxicity? Please add a table related to this.
Answer: Added as table.
Minor Concerns:
- Authors mentioned "A number of QTLs associated 28 with HM stress signals from various intermediary molecules have been reported to activate various transcription factors." but not enlisted any single one.
Answer: In order to make manuscript easier to read, the data regarding these QTLs has been added as table.
- The color scheme in the legends of Figure 1 is not clear. Please make it clearer.
Answer: Figure is further improved. More suggestions are appreciated.
- A study conducted in 2020 at eastern China examined HM level in urban soil. Considering the environmental and health risks" please provide a reference.
Answer: Reference added and improved. Li, Y., et al., Study on the risk of soil heavy metal pollution in typical developed cities in eastern China. Scientific reports, 2022. 12(1): p. 3855.
- in line 321: OsABCB24 gene it must be italic. please check the whole manuscript carefully.
Answer: Improved as suggested
- Add a separate conclusion section with solid recommendations.
Answer: The conclusion section is separated and added with more detailed information.

Round 2
Reviewer 2 Report
The authors have addressed all the comments and improved the manuscript significantly.